# Chinese Economic Growth Projections Based on Mixed Data of Carbon Emissions under the COVID-19 Pandemic

Rong Fu [1], Luze Xie [1], Tao Liu [2,*] [ID], Juan Huang [1] and Binbin Zheng [1]

[1] College of Economics, Hangzhou Dianzi University, Hangzhou 310018, China
[2] Department of Sociology, Hangzhou Dianzi University, Hangzhou 310018, China
* Correspondence: liutao@hdu.edu.cn

**Abstract:** Current research on carbon emissions and economic development has tended to apply more homogeneous low-frequency data to construct VAR models with impulse responses, ignoring some of the sample information in high-frequency data. This study constructs a MIDAS model to forecast GDP growth rate based on monthly carbon emission data and quarterly GDP data in the context of the COVID-19 pandemic. The results show that: (1) The MIDAS model has smaller RMSE than the VAR model in short-term forecasting, and provides more stable real-time forecasts and short-term forecasts of quarterly GDP growth rates, which can provide more accurate reference intervals; (2) China's future macroeconomic growth rate has recently declined due to the impact of the sudden epidemic, but the trend is generally optimistic. By improving urban planning and other methods, the authorities can achieve the two-carbon goal of carbon capping and carbon neutrality at an early date. In the context of the impact of COVID-19 on China's economic development, we need to strike a balance between ensuring stable economic growth and ecological protection, and build environmentally friendly cities, so as to achieve sustainable economic and ecological development and enhance human well-being.

**Keywords:** COVID-19; carbon dioxide emissions; sustainability; urban planning; environmentally friendly cities

## 1. Introduction

The threat of climate change due to increased global warming has been a major environmental issue in the past decades, and one of the main causes of global warming and climate instability is the rise in carbon dioxide emissions. Anthropogenic activities such as over-reliance on fossil fuel-based electricity and heat production, as well as industrial production and construction that burn fossil fuels, have contributed to the emission of $CO_2$ [1,2]. The large amount of energy consumed as a result of economic growth has led to large emissions of pollutants such as carbon dioxide, which have had a negative impact on urban ecology and public health [3] while exacerbating the deterioration of environmental quality [4]. The problem of carbon emission from the energy system mainly based on fossil energy is becoming more and more prominent, and the response to climate change has become a great challenge for human society.

Carbon dioxide emissions are chronically correlated with economic growth [5]. The majority of $CO_2$ emissions come from fossil fuel consumption, which is an important source of industry that is closely linked to economic development and growth. Therefore, the inextricable relationship between $CO_2$ emissions and economic growth is an important bridge between economic and environmental policies [6]. In fact, the increase of $CO_2$ emissions is the main threat to climate change, which is a major ongoing concern for both developing and developed countries. As the world's largest developing country, China is one of the fastest growing economies in the world. In recent decades, China has experienced unprecedented development, brought about by large-scale industrial development,

transformation of the service sector, agricultural mechanization, and development of the construction industry. These rapid developments are supported by a large amount of energy consumption, which leads to the continuous and rapid growth of energy consumption in China, and the level of energy consumption has been high, resulting in a significant increase in carbon emissions and a series of environmental pollution problems [7–12]. Since 2006, China has become the world's largest emitter of carbon emissions [13]. The long-standing "coal-based" basic energy supply pattern and the high intensity of industrial energy consumption have led to a "linkage" between carbon emissions and economic growth. Therefore, it is important to study the relationship between economic growth and $CO_2$ emissions for the implementation of relevant policies [14].

In recent years, a large number of scholarly studies have confirmed the existence of an inseparable relationship between carbon dioxide emissions and economic growth. In terms of research on the causal relationship between $CO_2$ emissions and economic growth, it is widely accepted that there is a bidirectional causal relationship between GDP and $CO_2$ emissions [15–18]. For example, Omri studied the causal relationship between $CO_2$ emissions, FDI and economic growth using a dynamic joint cubic equation panel data model, and the results provided evidence for a bidirectional causal relationship between FDI inflows and economic growth and between FDI and $CO_2$ emissions for all panels [19]. However, not all studies show an inverted U-shaped curve between these two variables. For example, Farhani and Ozturk provide direct evidence of a monotonic relationship between economic growth and carbon dioxide emissions [20]; the results of Saboori and Sulaiman show a significant non-linear relationship between carbon emissions and economic growth [21]; Heidari also confirmed a non-linear relationship between per capita $CO_2$ emissions, per capita energy consumption, and GDP [22]. Saidi examined the effect of energy consumption and carbon dioxide emissions on economic growth using a joint cubic equation model with panel data for 58 countries over the period 1990–2012, and the results showed that carbon dioxide emissions have a negative impact on economic growth [23]. Adams' results based on the Panel Pooled Mean Group-Autoregressive Distributed lag model show that economic growth contributes to $CO_2$ emissions [24]. Wang conducted an empirical analysis using panel data from 30 provincial levels in China, proving a positive relationship between economic growth targets and carbon emissions [25]. Zhao quantitatively analyzed the relationship between carbon emissions and economic development through Tapio decoupling model and LMDI decomposition model, and the research results showed that carbon emissions were positively correlated with economic development [26]. Despite many controversial and ambiguous results, the overall conclusion of these studies is that there is a close relationship between $CO_2$ emissions and economic growth. Hoa and Limskul presented a quantitative structural modeling perspective and policy analysis in an economic integration framework [27], and systematic estimation of the causal relationship between aggregate growth and $CO_2$ emissions, as well as for major developing countries in Asia. Therefore, accurate forecasting of economic development based on estimating the relationship between carbon emissions and economic growth is important because it can provide important references for policy makers to take preventive measures in urban planning, energy adjustment and other aspects, so as to promote sustainable economic and ecological development.

In the current COVID-19 environment, coronavirus disease has a significant impact on both carbon dioxide emissions and economic development [28]. Around the globe, countries and cities have taken drastic measures to try to stop the spread of the virus, which has brought major economic and transportation activities to a sudden halt. As a result, the temporary lockdown period contributed to a sharp drop in global daily carbon dioxide emissions [29]. For example, Cioca has demonstrated a 12% reduction in air pollution after lockdown in European urban areas, with a substantial decrease in all air pollutants [30]. Many studies predict that current $CO_2$ reductions will be temporary and that if the necessary measures are not taken, emissions will return to pre-COVID-19 levels as economic activity partially recovers [31–33]. Long-term monetary and fiscal policies therefore need to

be adjusted to accommodate green and healthy economies [34], while addressing climate change and the health crisis to promote well-being of the population [35].

However, the current research on carbon emissions and economic development tends to use low-frequency data of the same frequency to construct models, ignoring some of the sample information in high-frequency data and erasing the fluctuations of high-frequency data. Because traditional forecasting models require the same data frequencies, higher frequency daily and monthly data are usually converted to lower frequency quarterly data, such as averaging, bridging [36], and temporal aggregation [37]. However, these frequency conversion methods may result in the loss of a considerable portion of the information contained in high-frequency data, such as fluctuations in high-frequency data, thus reducing the efficiency of using sample information to some extent [38]. In order to solve the above problems of different frequencies, Ghysels et al. proposed a hybrid data sampling model [39] and conducted some exploration and application [40–42]. Compared with the traditional model, the mixed-frequency data sampling regression model (MIDAS) constructs the distribution lag polynomial [43] to balance the relationship between retaining valid information of high-frequency data and reducing the number of parameters to be estimated [44]. The MIDAS model can effectively use high-frequency data to improve the prediction accuracy of low-frequency variables and avoid a large amount of loss of sample information. Without the limitation of requiring the same frequency data to build the model, MIDAS model has great advantages in using high-frequency data to forecast quarterly GDP growth in real time [45–49].

The research objectives of this paper are as follows: firstly, based on monthly $CO_2$ emissions and quarterly GDP data of China from 2019 to 2021 in the context of COVID-19, MIDAS models with different weight functions will be established. Secondly, different forecasting models will be established after choosing the optimal lag order. Finally, according to the forecast results, the change trend of China's GDP in recent years will be analyzed, and the corresponding carbon emission reduction countermeasures will be put forward. This will help China meet the challenge of maintaining its economic growth rate until 2030, reaching peak $CO_2$ emissions around 2030 [50] and reducing $CO_2$ emissions per unit of GDP by 60–65% compared to 2005 [51], as promised to the international community.

The rest of the paper is organized as follows: Section 2 presents the modeling approach used in this paper. Section 3 provides an empirical analysis of GDP growth forecasts, which demonstrates the validity of the MIDAS model proposed in this study. Section 4 discusses the differences between the study results and the latest literature. Section 5 discusses the theoretical contributions, policy implications, limitations, and future research perspectives of the study.

## 2. Materials and Methods

### 2.1. Data

Currently, China, as the world's largest coal-fired country in terms of total carbon emissions, uses energy consumption to drive its rapid economic development. Therefore, this paper examines the impact of monthly $CO_2$ emissions on quarterly economic growth and makes accurate forecasts of economic growth. The sample data covers monthly $CO_2$ emissions and corresponding quarterly GDP data from January 2019 to June 2022 during the COVID-19 pandemic. The data used for the high-frequency monthly variable carbon emissions are obtained from a global real-time carbon data site called carbon monitor, and the data used for the low-frequency quarterly variable GDP are obtained from China's National Bureau of Statistics. The study of the relationship between carbon emissions and economic growth is important for China to achieve the goal of "double carbon" and high-quality development.

The growth rate of the variables was considered in the empirical analysis to eliminate heteroskedasticity, so the growth rate was defined as follows:

$$growth_{it} = \ln\left(\frac{value_{it}}{value_{it-1}}\right) \times 100 \tag{1}$$

where $growth_{it}$ denotes the growth rate of indicator $i$ at time $t$, and indicator $i$ denotes monthly $CO_2$ emissions or real quarterly GDP.

Figure 1 shows the trends of $CO_2$ emissions and quarterly GDP and their growth rates. It is clear from the figure that monthly $CO_2$ emissions have a similar trend to quarterly GDP. During the study period, both $CO_2$ emissions and GDP fluctuated significantly, but their trends were roughly consistent. Their fluctuations probably stem from the effects of COVID-19, where measures such as closures and shutdowns caused them to decrease sharply and then rise again when work resumed. Their turning points basically coincide, and the data changes were relatively consistent. Fluctuations in carbon dioxide emissions led to corresponding fluctuations in GDP. Therefore, this similarity provides a basis for using monthly carbon dioxide emissions to forecast quarterly GDP.

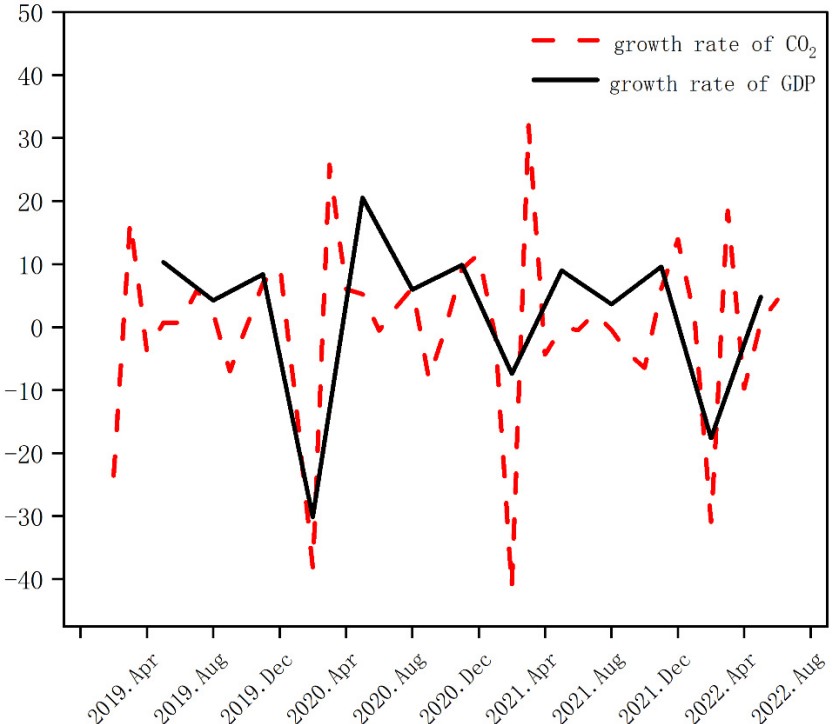

**Figure 1.** Graph of economic growth and carbon emission growth rate.

*2.2. The MIDAS Regression Model*

The MIDAS model allows variables of different frequencies to be constructed in the same model and is used to study the effect of changes in high-frequency variables on low-frequency variables. The basic MIDAS (m, K) can be simply expressed as

$$Y_t = \beta_0 + \beta_1 B\left(L^{1/m};\theta\right) X_t^{(m)} + \varepsilon_t^{(m)} \tag{2}$$

Equation (2) is the underlying one-equation model, where $X_t^{(m)}$ is the high-frequency explanatory variable. $Y_t$ is the low-frequency explanatory variable, and m is the frequency multiplicative difference between the explanatory variable and the explanatory variable, which is set in this paper $X_t^{(m)}$ is a monthly variable and $Y_t$ is the quarterly variable, then the value of $m$ is 3. $B\left(L^{1/m};\theta\right)$ is the lagged weight polynomial, which is the combination of weight function and lag operator, and can be written as $\sum_{k=0}^{K} \omega(k;\theta) L^{k/m}$, where $K$ is the highest lag order of the high-frequency explanatory variables. $\omega(k;\theta)$ is the weight function, and $L^{k/m}$ is the lag operator, and $X_t^{(m)}$ is the combination of $X_{t-k/m}^{(m)}$, $k = 0, 1, \cdots, K$.

Since the data release has a time lag, for example, the quarterly GDP is usually released in the middle and end of the first month of the next quarter, the introduction of the MIDAS

model with h-step forward forecasting can make full use of the published high-frequency data to forecast the low-frequency data. Compared with the same frequency model, the MIDAS model can correct its forecast according to the newly released data, which improves the forecast accuracy and solves the time lag problem of data publication. The h-step forward forecasting model is abbreviated as MIDAS (*m*, *K*, *h*), which can be written as

$$Y_t = \beta_0 + \beta_1 B\left(L^{1/m}; \theta\right) X^{(m)}_{t-h/m} + \varepsilon^{(m)}_t \tag{3}$$

In Equation (3), $B\left(L^{1/m}; \theta\right)$ is the same as above and is still $\sum_{k=0}^{K} \omega(k;\theta) L^{k/m}$. The difference is that when the lag operator is combined with $X_t^{(m)}$, it is $L^{k/m} X^{(m)}_{t-h/m} = X^{(m)}_{t-k/m-h/m} = X^{(m)}_{t-(k+h)/m}$. In this case, $K + h$ is the highest lag order calculated at the high-frequency of the high-frequency explanatory variable. When $h = 1$, the information of the first two months of high-frequency data is available in the current quarter data to be forecasted, and the data of the first two months can be used to forecast the information at the end of the quarter; when $h = 2$, only the information of the first month is known in the current quarter data to be forecasted, and the data of the first month can be used to forecast the information at the end of the quarter; when $h > 3$, it means that the regression equation can make out-of-quarter forecast based on the existing data. For example, it can forecast the information of the next quarter using the data of the previous period and revise its forecast using the latest high-frequency data.

One of the core elements of the MIDAS model is the setting of the weight function. Introducing the weight function in the model not only reduces the influence of noise in high-frequency data, but also allows the analysis of the structure of high-frequency data. Using the characteristics of the function itself, the numerous estimated parameters of the high-frequency explanatory variables in the model are represented by the function, making it possible to introduce high-frequency variables directly in the model. The weight functions commonly used in the MIDAS model in empirical studies are Beta polynomial function, non-zero Beta polynomial function, Almon function, and Exponential Almon function [52,53].

The Beta polynomial function is a representation of the weight function of the MIDAS model using probability density functions from the family of Beta distributions, mainly from the fact that a rich form of the probability density function of the family of Beta distributions can be represented by using only two parameters [39]:

$$\omega(k; \theta_1, \theta_2) = \frac{f(k/K; \theta_1, \theta_2)}{\sum_{k=0}^{K} f(k/K; \theta_1, \theta_2)} \tag{4}$$

In Equation (4), $f(X; a, b) = \frac{X^{a-1}(1-X)^{b-1}\Gamma(a+b)}{\Gamma(a)\Gamma(b)}$ and $\Gamma(a) = \int_0^\infty e^{-x} x^{a-1} dx$. $f(X; a, b)$ takes different forms as the parameters change. When $a > 1$, $b > 1$, $f(X; a, b)$ is a sine function and at $x = \frac{a-1}{a+b-2}$ obtains its maximum value at this time. When $a < 1$, $b < 1$, $f(X; a, b)$ is a U-shaped function and it obtains the minimum value in $x = \frac{1-a}{1-a-b}$. When $a > 1$, $b \leq 1$, $f(X; a, b)$ is a strictly decreasing function. Therefore, the weight function $\omega(k; \theta_1, \theta_2)$ will change with the value of the parameter $\theta_1$ and $\theta_2$.

A non-zero Beta polynomial function is defined by the case where none of the parameters in the Beta weight function is zero, as follows

$$\omega(k; \theta_1, \theta_2) = \frac{f(k/K; \theta_1, \theta_2)}{\sum_{k=0}^{K} f(k/K; \theta_1, \theta_2)} + \theta_3 \tag{5}$$

As the name implies, $\theta$ does not equal 0 in non-zero Beta polynomial functions. It is widely used when $\theta_1 = 1$, $\theta_2$ and $\theta_3$ are not 0. In Equation (4) $\theta$ can be 0, which can be obtained from Equation (5) $\theta_3 = 0$. This type of weighting function is often used

in financial markets and combined with macroeconomic areas, such as the use of high-frequency financial variables to predict low-frequency macroeconomic variables [54].

The Almon function is mainly derived from the van der Munn matrix in the distribution lag model, and the main idea is to approximate a distribution by a polynomial, as defined below:

$$\omega(k;\theta_0,\theta_1,\theta_2\cdots,\theta_P) = \theta_0 + \theta_1 k + \theta_2 k^2 + \cdots + \theta_P k^P \tag{6}$$

When expanded, it can be written as

$$\begin{pmatrix} \omega_0 \\ \omega_1 \\ \omega_2 \\ \omega_2 \\ \vdots \\ \omega_k \end{pmatrix} = \begin{pmatrix} 1 & 0 & 0 & \cdots & 0 \\ 1 & 1 & 1^2 & \cdots & 1^P \\ 1 & 2 & 2^2 & \cdots & 2^P \\ \vdots & \vdots & \vdots & \ddots & \vdots \\ 1 & k & k^2 & \cdots & k^P \end{pmatrix} \begin{pmatrix} \theta_0 \\ \theta_1 \\ \theta_2 \\ \vdots \\ \theta_P \end{pmatrix} \tag{7}$$

In order to reduce the parameters to be estimated, in general one would make $P = 3$.

The exponential Almon function is developed from the Almon hysteresis polynomial proposed by the distributed hysteresis model, that is, from the general form of the Almon function:

$$\omega(k;\theta_1,\theta_2\cdots,\theta_P) = \frac{\exp(\theta_1 k + \theta_2 k^2 + \cdots + \theta_P k^P)}{\sum_{k=1}^{K} \exp(\theta_1 k + \theta_2 k^2 + \cdots + \theta_P k^P)} \tag{8}$$

To avoid the problem of too many parameters to be estimated, parameter $P = 2$ is generally set. At this point there is a general constraint of $\theta_1 \leq 300, \theta_2 < 0$. It has become a popular option because it is flexible enough to simulate different weighted shapes of lag coefficients.

In addition to the above models with weight functions as constraints, some scholars have also proposed the unconstrained MIDAS model [55], referred to as the U-MIDAS model. As the name implies, this model does not need to impose constraints on the lag polynomial and can be regressed directly by the least square method. According to the modeling mechanism of the traditional distributed lag model, the unconstrained MIDAS model is obtained by removing the weight function before the high-frequency variables:

$$Y_t = \beta_0 + \beta_1 B\left(L^{1/m}\right) X_t^{(m)} + \varepsilon_t^{(m)} \tag{9}$$

There is no weight function in Equation (9), and it is necessary to estimate the parameters of the lagged term of each high-frequency explanatory variable, and the other variables have the same meaning as in Equation (2). The model can be used to make predictions for low-frequency variables when the difference between the frequencies of high-frequency and low-frequency variables is small and the estimated parameters are not large.

## 3. Results

The choice of weight function and lag order is very important. In practice, different forms of weight function are introduced into the MIDAS regression model, and the optimal lag order of the weight function is selected according to the characteristics of high frequency data fluctuation and the specific value of the model prediction accuracy index. The processed carbon emission data are recorded as $\mathrm{Dlog}(CO_2)$ and the GDP data are recorded as $\mathrm{Dlog}(GDP)$.

### 3.1. Correlation Analysis of Carbon Emissions and Economic Growth

In order to test whether there is a correlation between carbon dioxide emissions and economic growth, this study transformed carbon emissions into quarterly data and then conducted Granger causality tests, and obtained the results shown in Table 1.

**Table 1.** Granger causality test between carbon emissions and economic growth.

| Dependent Variable | Excluded | Chi-sq | df | Prob. |
|---|---|---|---|---|
| $Dlog(CO_2)$ | $Dlog(GDP)$ | 11.9432 | 3 | 0.0076 |
| $Dlog(GDP)$ | $Dlog(CO_2)$ | 43.7252 | 3 | 0.0000 |

As can be seen from Table 1, the *p*-values of the Granger causality tests between $Dlog(CO_2)$ and $Dlog(GDP)$ are less than 0.05, indicating that the two are mutually influencing and constraining, and a vector autoregressive model (VAR) can be considered.

The values of LR, FPE, AIC, SC, and HQ were considered together and the model lag order of VAR was set as 3. The results of the parameter estimation of the final constructed VAR model are shown in the Equation (10).

$$\begin{bmatrix} Dlog(CO_2) \\ Dlog(GDP) \end{bmatrix} = \begin{bmatrix} -0.62 & 0.24 \\ 0.19 & -0.26 \end{bmatrix} \times \begin{bmatrix} Dlog(CO_2)(-1) \\ Dlog(GDP)(-1) \end{bmatrix} + \begin{bmatrix} -0.31 & -0.09 \\ -1.16 & 0.46 \end{bmatrix}$$
$$\times \begin{bmatrix} Dlog(CO_2)(-2) \\ Dlog(GDP)(-2) \end{bmatrix} + \begin{bmatrix} 1.94 & -1.59 \\ 3.98 & -2.93 \end{bmatrix} \times \begin{bmatrix} Dlog(CO_2)(-3) \\ Dlog(GDP)(-3) \end{bmatrix} + \begin{bmatrix} 3.26 \\ 6.56 \end{bmatrix} \tag{10}$$

From Equation (10), it can be seen that $Dlog(GDP)$ with lag 1 has a promoting effect on $Dlog(CO_2)$, $Dlog(GDP)$ with lag 2 and lag 3 has a suppressing effect on $Dlog(CO_2)$; $Dlog(CO_2)$ with lag 1 and lag 3 has a promoting effect on $Dlog(GDP)$, $Dlog(CO_2)$ with lag 2 has a suppressing effect on $Dlog(GDP)$. Therefore, it is reasonable to use $Dlog(CO_2)$ to predict $Dlog(GDP)$.

### 3.2. The MIDAS Model

Commonly used weighting functions are exponential Almon function, Almon function, Beta polynomial function, and non-zero Beta polynomial function. The variables are constructed as full sample mixed frequency data models, and different models are constructed by combining the possible weight functions and lag structures of the models, and the results are shown in Table 2.

**Table 2.** Model fitting of different weight functions at different lag periods.

| Lagging Period | Weight Functions | AIC | BIC | Convergence |
|---|---|---|---|---|
| 1 | neAlmon | 97.5931 | 99.5327 | 0 |
| 2 | neAlmon | 97.5931 | 99.5327 | 0 |
| 3 | neAlmon | 97.5931 | 99.5327 | 0 |
| 4 | neAlmon | 97.5931 | 99.5327 | 0 |
| 5 | neAlmon | 97.5931 | 99.5327 | 0 |
| 6 | neAlmon | 97.5931 | 99.5327 | 0 |
| 2 | Almon | 85.5160 | 87.9406 | 0 |
| 3 | Almon | **76.0655** | **78.4901** | 0 |
| 4 | Almon | 80.4132 | 82.8378 | 0 |
| 5 | Almon | 87.5142 | 89.9387 | 0 |
| 6 | Almon | 100.4989 | 102.9234 | 0 |
| 2 | nBeta | 99.5931 | 102.0176 | 0 |
| 3 | nBeta | 99.5931 | 102.0176 | 0 |
| 4 | nBeta | 99.5931 | 102.0176 | 0 |
| 5 | nBeta | 99.5931 | 102.0176 | 0 |
| 6 | nBeta | **84.2460** | **86.6705** | 0 |

**Table 2.** *Cont.*

| Lagging Period | Weight Functions | AIC | BIC | Convergence |
|---|---|---|---|---|
| 3 | nBetaMT | **91.7440** | **94.6535** | 0 |
| 4 | nBetaMT | 99.9983 | 102.9078 | 0 |
| 5 | nBetaMT | 100.3708 | 103.2802 | 0 |
| 6 | nBetaMT | 105.4704 | 108.3798 | 0 |

\* Note: "**Bond**" is the minimum value of AIC or BIC under this weight function.

From Table 2 and Figure 2, it is known that the AIC and BIC values of the exponential Almon function do not vary with lags. The Almon function and the non-zero Beta polynomial function reach the minimum of the AIC and BIC values at lag K = 3, and the Beta polynomial function reaches the minimum of the AIC and BIC values at lag K = 6. In addition, the convergence values of all models are 0, which means that the constructed models converge. Since the Almon function and the non-zero Beta polynomial function have the maximum AIC and BIC at lag K = 6, and the minimum values of AIC and BIC are obtained at lag K = 3 of the Almon function, the lag order K = 3 is chosen.

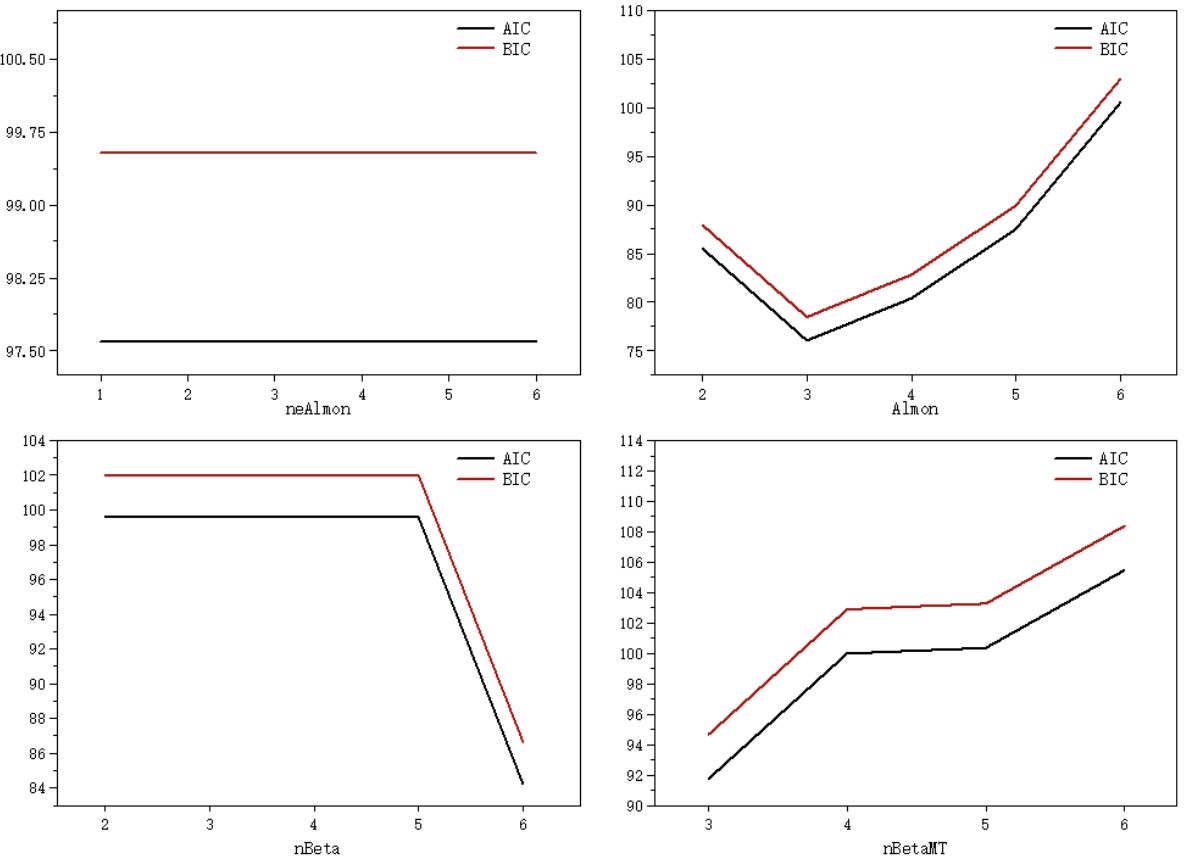

**Figure 2.** Model fitting diagram of five weight functions changing with different lag periods.

With the optimal high frequency lag value K = 3, to examine the merits of the MIDAS model, the MIDAS models with different constraint functions were constructed to predict GDP growth rate by choosing the first quarter of 2019 to the second quarter of 2021 as the training set, and the third quarter of 2021 to the second quarter of 2022 as the test set. Table 3 gives the errors of the sample forecasts of GDP growth rate.

According to Figure 3 and Table 3, it can be seen that the out-of-sample errors of the Beta polynomial function and the exponential Almon function are the same, and the out-of-sample errors of Almon function are the smallest for both MSE, MAPE and MASE.

Therefore, the model constructed using the Almon function has the best predictive ability for the future, and the Almon model with lag K = 3 can be selected by comparison.

**Table 3.** The sample prediction error of different weight functions when the lag is 3.

| Lagging Period | Weight Functions | MSE.out | MAPE.out | MASE.out |
| --- | --- | --- | --- | --- |
| | nBeta | 87.4890 | 102.3061 | 0.4352 |
| | nBetaMT | 63.4897 | 78.7915 | 0.3708 |
| 3 | unconstrained | 57.5514 | 79.9784 | 0.3608 |
| | neAlmon | 87.4890 | 102.3061 | 0.4352 |
| | Almonp | **27.6672** | **49.3636** | **0.2261** |

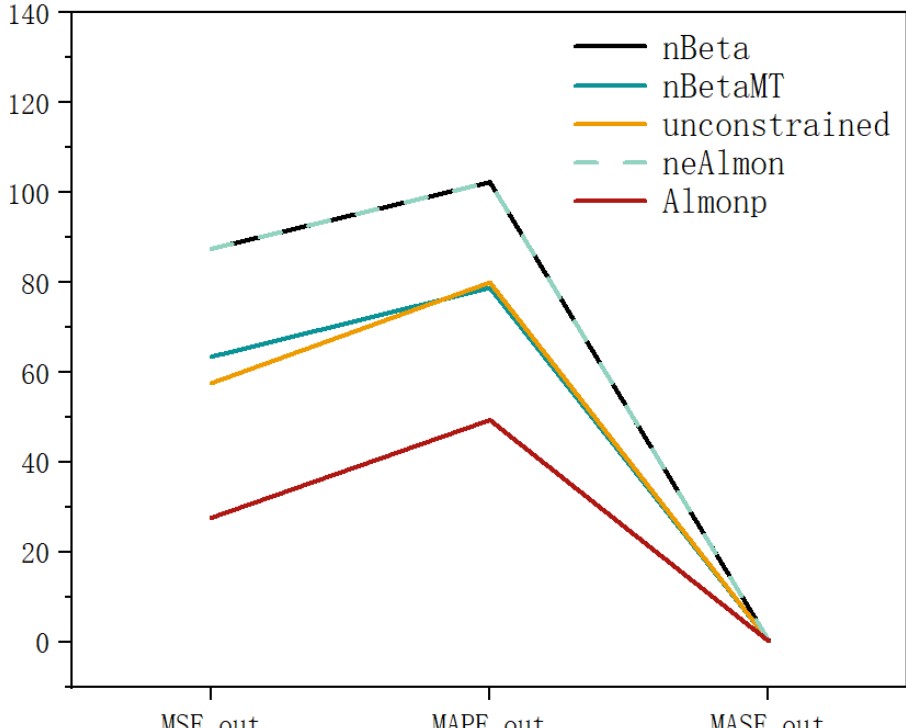

**Figure 3.** Error comparison of different weight functions when lag is 3.

### 3.3. Model Comparison and Prejections

The constructed MIDAS model was subjected to out-of-sample prediction and the constructed VAR model was fitted in-sample for model comparison, and the results obtained are shown in Table 4.

As can be seen from Table 4 and Figure 4, at forecast steps less than 6, the RMSE of the MIDAS model is smaller than that of the in-sample fitted VAR model even for out-of-sample forecasts, reflecting the strong short-term forecasting ability. In addition to this, the MIDAS model performs worse and worse out-of-sample prediction results as the prediction base period h gradually increases, and the comparative advantage of the model is small or almost nonexistent. This is because as the step value increases, the corresponding amount of recent information decreases and the forecasting effect becomes worse. At the same time, it shows that the economy has the so-called sticky, and the economic situation of the next period is greatly affected by the economic level of the previous period. Thus, when forecasting the actual GDP growth rate further away from the forecast interval, it does not show better estimation effect than the VAR model at the same frequency. However, in general, the MIDAS model has a satisfactory accuracy for short-term forecasts of economic growth.

**Table 4.** Error comparison between the MIDAS model and the VAR model.

| | MIDAS | | | | VAR | | | | rRMSE |
|---|---|---|---|---|---|---|---|---|---|
| h | Time | Forecasted Value | RMSE | MAE | Time | Fitted Value | RMSE | MAE | |
| 1 | July 2021 | 1.23 | | | | | | | |
| 2 | August 2021 | 1.43 | 0.95 | 0.94 | Third quarter of 2021 | 6.49 | 2.83 | 2.83 | 0.34 |
| 3 | September 2021 | −0.48 | | | | | | | |
| 4 | October 2021 | −3.32 | | | Fourth quarter of 2021 | 4.46 | 4.16 | 3.99 | 0.78 |
| 5 | November 2021 | −0.79 | 3.24 | 2.24 | | | | | |
| 6 | December 2021 | 6.81 | | | | | | | |
| 7 | January 2022 | 8.21 | | | First quarter of 2022 | −17.62 | 3.39 | 2.66 | 1.41 |
| 8 | February 2022 | −6.60 | 4.77 | 3.69 | | | | | |
| 9 | March 2022 | −6.15 | | | | | | | |
| 10 | April 2022 | 0.02 | | | Second quarter of 2022 | 10.74 | 4.19 | 3.49 | 1.28 |
| 11 | May 2022 | −0.06 | 5.35 | 4.46 | | | | | |
| 12 | June 2022 | 0.82 | | | | | | | |

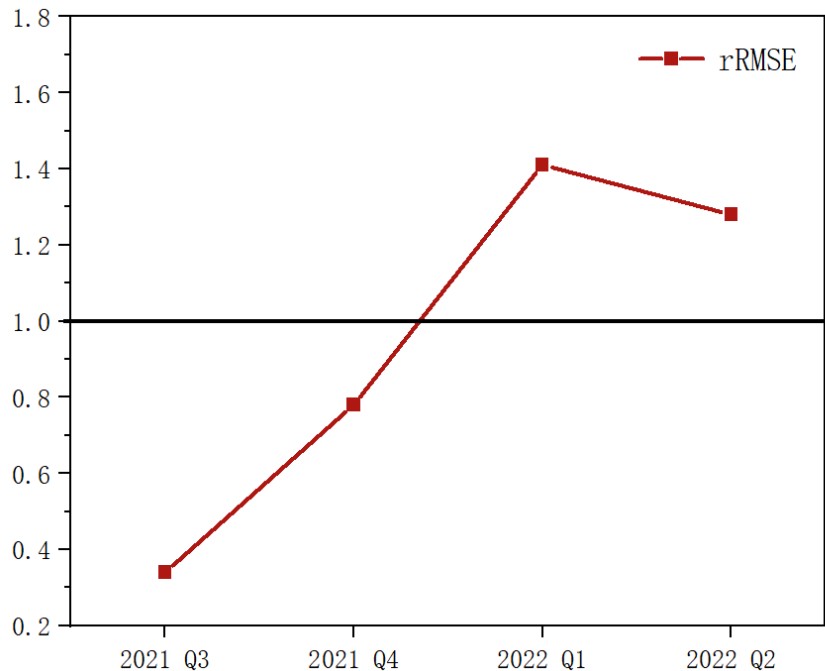

**Figure 4.** rRMSE comparison between the MIDAS model and the VAR model.

Projections of GDP growth based on carbon emissions data yielded the results shown in Table 5. From Table 5 and Figure 5 it can be seen that the forecast shows a general upward trend for the Chinese economy over the next three months, but the growth rate may be affected by the sudden outbreak of the COVID-19 epidemic and will decline. In the short term, the sudden outbreak of the COVID-19 may lead to a decline in the domestic economy, but in the medium and long term, the negative impact on economic development is limited. Based on reasonable expectations of China's economic growth in the short and medium term, and in relation to the findings related to the total and structure of society-wide carbon emissions, it can be tentatively judged that it is reasonable and feasible to maintain a sustained decline in domestic carbon emissions intensity, provided that pre-conditions remain unchanged. Due to the predictive advantage of the MIDAS model in the short term, the model's predictions can be revised and adjusted after the release of the latest data.

**Table 5.** A three-step forecast of future economic growth.

| h | Time | Forecasted Value |
|---|------|------------------|
| 1 | July 2022 | 1.7315 |
| 2 | August 2022 | 1.4050 |
| 3 | September 2022 | −0.8291 |

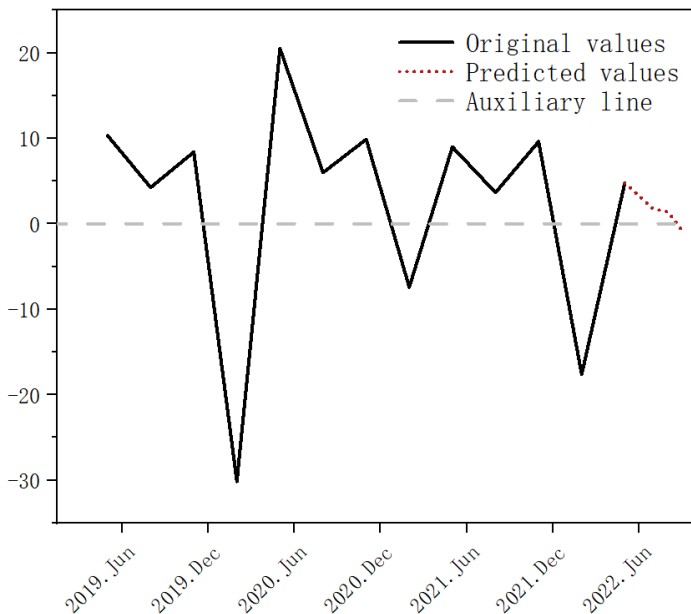

**Figure 5.** Forecast trend of economic growth.

## 4. Discussion

Recent studies attest to the rapid spread of COVID-19 around the world, with countries locking down cities, restricting travel and implementing stringent security measures. These have an impact on the environment and carbon emissions [56–58], while holding back economic development [59–61]. Therefore, we want to study the impact of carbon emissions on the economy so that policymakers can make appropriate policies to strike a balance between economic development and environmental protection.

This study is the first to use the MIADS regression model to analyze the impact of carbon emissions on economic growth in the context of COVID-19. The Granger causality test proves the rationality of using carbon emission to forecast economic growth. The forecasting accuracy of the MIDAS model is demonstrated by comparing it with the VAR model. Finally, the forward three step forecast reveals the trend of future economic changes. The results of this study suggest that China's future economic development has been affected by the sudden epidemic, resulting in a recent decline in economic growth rate, but overall, the Chinese economy is still on an upward trend. In general, Jardet used the frequency mixing model to predict the world economy [62], and this study added the screening of different constraint functions and lag orders on the basis of his model, and reached a similar conclusion, that is, high-frequency data can help track economic turning points in real time. Xu found that the prediction accuracy of MIDAS model was higher than that of autoregressive Distributed lag (ARDL) model through the study of carbon emission prediction [63], which was similar to this study. Different from Xu, this study predicts economic growth based on the data of carbon emissions under the COVID-19 pandemic and finds that the MIDAS model predicts with higher accuracy than the VAR model. Empirical research shows that the development trend of economic growth rate and carbon emission growth rate shows consistency and phase characteristics, which is consistent with Zhao's findings [26]. However, Zhao analyzed the relationship between carbon emissions and economic development from 2009 to 2019. This study can be said to

be an extended analysis of the relationship between 2019 and 2022, while using the MIDAS model to make the short-term prediction more accurate. In terms of economic forecast results under the background of COVID-19, Zhao used the conditional Markov chain model to forecast China's GDP growth rate from the fourth quarter of 2021 to the third quarter of 2025 [64]. The main findings are that COVID-19 has an impact on China's economic growth, China's economic fluctuations are large, and economic growth differences are large. However, China's economic growth is likely to stabilize gradually and is likely to grow at a medium-high speed with high quality during the forecast period, similar to the forecast results of this study. In this study, the sudden epidemic was further considered, that is, the monthly data was used to predict the quarterly data. With the emergence of the sudden epidemic, the economic growth rate would suffer a certain decline, but it would rise after the lockdown exposure, showing obvious volatility.

## 5. Conclusions

### 5.1. Theoretical Contributions

Environmental problems caused by carbon emissions have become a topic of increasing concern, and a large amount of carbon emissions is behind the rapid economic development, so it is necessary to study the relationship between carbon emissions and economic development. In the latest studies, the relationship between economic development and carbon emissions is generally explored by decompressing the influencing factors of carbon emissions, or directly constructing an econometric model of carbon emissions and economic development to explore the direct relationship between them. However, they all use the same frequency data. In this study, for the first time, the mixed-frequency data sampling regression method was used to analyze the impact of carbon dioxide emissions on economic growth under the background of COVID-19 pandemic. The MIDAS model balances the relationship between retaining valid information of high-frequency data and reducing the number of parameters to be estimated by constructing distribution lag polynomials, thus effectively using high-frequency data to improve the prediction accuracy of low-frequency variables and avoiding a large loss of sample information. At the same time, the accuracy of the MIDAS model is verified by comparing the VAR model, which can better reflect the impact of short-term and sudden epidemics on economic growth.

The purpose of this study is to investigate the impact of $CO_2$ emissions on macroeconomic fluctuations under the COVID-19 in China by using the MIDAS model. The MIDAS (m, K) model for estimating the real GDP growth rate is constructed from $CO_2$ emissions, the variables are constructed as a full-sample MIADS model, the possible weight functions and lag structures of the model are combined to construct different models to determine the lag order K of the model, and the optimal weight function is further selected through the sample forecast error. On this basis, the sample estimation results are used to obtain the forecasted GDP growth rate and the root mean square residual ratio for comparison with the benchmark model VAR. Finally, the future GDP is projected based on the latest carbon emission data, and corresponding recommendations are made to achieve sustainable development.

### 5.2. Policy Implications

Through the above empirical study, this study finally draws the following conclusions:

Firstly, compared with the benchmark model VAR, the MIDAS model has a smaller mean-squared forecast error in general, and has a comparative advantage in real-time forecasting and short-term forecasting, avoiding the time lag in the release of economic data that prevents timely and accurate judgments on the current macroeconomic state and macroeconomic trends, and improving the timeliness of macroeconomic forecasts and the accuracy of short-term forecasts.

Secondly, when the MIDAS model forecasts the real GDP growth rate, the best estimates as well as the forecast results are based on the short-term impact. In the short term, our economy is affected by the impact of the sudden epidemic. Therefore, we need to pay

attention to the economic trends, through the use of the MIDAS models in time for forecast monitoring, and thus modify the previous data.

Third, China's current and future ecological and environmental conditions remain optimistic under the influence of carbon emissions. The authorities concerned can promote the optimization and upgrading of industrial structure and the construction of environmentally friendly cities by improving urban planning and other methods, so as to achieve the dual carbon goals of carbon capping and carbon neutrality at an early date.

Based on the above findings and the trends in China's economic growth and carbon emissions, it is clear that the issue of how to control carbon emissions and maintain economic growth is a persistent and hidden problem. Reducing carbon emissions to control environmental pollution is bound to curb the rapid growth of the economy, which can be verified from the development path of the high carbon economic model that China has been pursuing. The previous development model achieved rapid economic development with high energy consumption and high pollution, which would not only increase the pressure of carbon emissions and aggravate the degree of environmental pollution, but also go against the sustainable development of economy. In addition, the environmental problems associated with carbon emissions, such as land drought, increased desertification and ocean acidification, can limit sustainable economic development in various ways. Therefore, a win–win situation of reducing carbon emissions and maintaining economic development can only be achieved by formulating appropriate carbon reduction policies, such as implementing carbon neutral concepts in urban planning, and building environmentally friendly cities, livable urban communities, and zero carbon buildings. From the empirical evidence it is clear that China's economy will continue to grow in the long term, so it is possible to invest more in carbon emission reduction, improve the institutions and mechanisms related to carbon emission reduction, and accelerate the innovation of energy technology. China's current economic development is affected by the epidemic, and it is important to strike a balance between ensuring stable economic growth and balancing ecological and environmental protection to bring maximum benefits to social development and enhance human well-being. Therefore, it is important to study the coupled and coordinated relationship between carbon emissions and economic growth to promote sustainable economic and ecological development.

### 5.3. Limitations

The research in this study has certain limitations. Since carbon dioxide mainly comes from the combustion of fossil fuels such as coal, oil, and natural gas, and energy is the main driving force for economic growth, carbon emissions and economic development are closely related to energy consumption, which leads to carbon emissions and economic growth. Therefore, the research results of this paper are worthy of further promotion of adding energy consumption as an independent variable into the prediction model. In addition, the MIDAS model is suitable for short-term forecasting, so timely data updates are needed to obtain more accurate results.

### 5.4. Future Research Perspectives

In view of the shortcomings of the research, the article can be improved from the following aspects if the conditions allow. First, a carbon emission index system can be constructed to incorporate energy consumption and other factors into the model to make the model more comprehensive. Second, the model can be used to further predict the short to medium term economic development, and update the forecast data in real time to improve the forecast accuracy. Third, a dummy variable can be set for the sudden epidemic to more comprehensively consider its impact on economic development, so as to timely adjust relevant policies. Subsequent research can be carried out according to the above aspects, so as to explore more profound laws of economic change and predict possible development trends.

**Author Contributions:** Conceptualization, R.F. and L.X.; methodology, R.F.; software, L.X.; validation, B.Z. and R.F.; formal analysis, T.L.; investigation, J.H.; resources, R.F.; data curation, L.X.; writing—original draft preparation, L.X.; writing—review and editing, R.F.; visualization, B.Z.; supervision, J.H.; project administration, T.L.; funding acquisition, R.F. All authors have read and agreed to the published version of the manuscript.

**Funding:** This research was funded by the National Social Science Fund of China (Grant No.20BTJ005).

**Institutional Review Board Statement:** Not applicable.

**Informed Consent Statement:** Not applicable.

**Data Availability Statement:** Data sources are contained within the article.

**Acknowledgments:** We would like to thank the reviewers for their valuable comments and efforts.

**Conflicts of Interest:** The authors declare no conflict of interest.

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
