# Peer review of "Chinese Economic Growth Projections Based on Mixed Data of Carbon Emissions under the COVID-19 Pandemic"

_sustainability, doi:10.3390/su142416762_

Round 1

Reviewer 1 Report

I liked the paper. The topic addressed is very interesting.

The main of the research is to improve the VAR models with impulse responses that ignore some of the sample information in high-frequency data.  The topic of the relationship between carbon emissions and economic development is part of the countries' agenda to reduce the footprint of climate change and to grow sustainable. Compared with other published material, the authors construct a MIDAS model to forecast GDP growth rate. the MIDAS model has smaller root mean square residuals than the general base model, and provides more stable real-time forecasts and short-term predictions of quarterly GDP growth rate, which can provide more accurate reference intervals. The methodology is appropriate for the research.   The conclusions are consistent with the arguments presented. By improving urban planning and other methods, the authorities can achieve the two-carbon goal of carbon capping and carbon neutrality at an early date. The bibliography is appropriate to the research topic and the number of research sources can reach the knowledge of the research stage.

• Very well documented paper and actual references;
• The organization of the paper;
• The authors have used an adequate model for this type of research;
• The results are correlated with previous studies in the field;
• The authors highlighted their contribution to the literature.

Author Response

Response to Reviewer 1 Comments

We thank the reviewers for their suggestions on this study. We have made the following main changes after receiving the review comments:

  1. Updated the latest data and redid the empirical analysis.
  2. Added Granger causality tests to demonstrate the correlation between carbon emissions and economic growth.
  3. Constructed a VAR model and used it as the base model for comparative forecasting.
  4. The number of forecasting periods is changed to 3 periods to obtain more accurate forecasting data.

Once again, we thank the reviewers for their efforts in this study.

Reviewer 2 Report

This study mainly applied a MIDAS model constructed by the authors to forecast GDP growth rate based on monthly carbon emission data and quarterly GDP data in the context of the COVID-19 pandemic from January 2019 to December 2021. This study acquire some relatively valuable conclusion and results,     gives some useful policy suggestions including ecological protection, and build environmentally friendly cities, and the like. This paper has a certain practical significance, and can enrich the research on relevant methods of economic growth prediction to a certain extent.  nevertheless, I have some suggestions or queries to discuss with the author of this study.  

The suggestions or questions are as follows:

(1)Is it scientific to predict economic growth based on the small sample data during the epidemic period? Even if monthly and quarterly data are used, the sample size is still limited. Could the authors give concerned reasonable explanation?

(2) During the epidemic period, national policies, scientific and technological investment and advancement, labor force, capital investment and other factors have been significantly adjusted, and the probability of various factors in the subsequent COVID-19 pandemic period will change. Is it reasonable to predict the economic growth in the later period based on the performance during the COVID-19 pandemic period? Please give the explanation in the paper.

(3)Why didn't the author adopt one or more mainstream approach to predict economic growth? It is suggested that the main economic growth forecasting methods and empirical studies at home and abroad be summarized.

(4)Please explain the advantages of MIDAS and the forecasting idea adopted in this paper compared with other economic growth forecasting methods!

(5) There is a close relationship between carbon dioxide emissions and economic growth, which is generally a conclusion drawn in a long-term study. Some studies also point out that there is a nonlinear relationship between the two aspects. What is the basis for the author to believe that there must be a close relationship between the carbon dioxide emissions and economic growth? In addition to the research of some domestic and foreign authors, has the author of this manuscript conducted a similar pre-study? Please provide additional information on this point.

(6)could the author explain that there is an objective correlation between CO2 emissions and quarterly GDP based on the data simulation of 36 months in a three-year period in China? And now the epidemic situation is still continuing, it is suggested that the author update the data.

(7)Why does the author not consider including the data before the epidemic? Why doesn't the author consider the monthly economic growth data? Please provide additional information on this two points.

(8) In the article, mainly discusses how to use monthly carbon dioxide emissions to forecast quarterly GDP. However, why does the author not consider that GDP fluctuations may cause changes in CO2 emissions? It is suggested to consider the two-way influence between the two.

(9) It is suggested to include Chinese in the title.

(10) In addition, the change of CO2 emissions may also cause many environmental problems. To what extent will this affect economic growth? This is also a question to be discussed. It is suggested to consider it.

Author Response

Response to Reviewer 2 Comments

We thank the reviewers for their suggestions on this study. We have made the following main changes after receiving the review comments:

  1. Updated the latest data and redid the empirical analysis.
  2. Added Granger causality tests to demonstrate the correlation between carbon emissions and economic growth.
  3. Constructed a VAR model and used it as the base model for comparative forecasting.
  4. The number of forecasting periods is changed to 3 periods to obtain more accurate forecasting data.

In response to your comments, we have made the following specific changes:

(1) Is it scientific to predict economic growth based on the small sample data during the epidemic period? Even if monthly and quarterly data are used, the sample size is still limited. Could the authors give concerned reasonable explanation?

Q1: Thanks for the reviewer's suggestion. This dataset has 36 data from January 2019 to December 2021, with a sample size greater than 30, which is sufficient to ensure the scientific validity of statistical inference. And based on the Q6, we supplemented the dataset to June 2022. The answer to this question is marked in line 148.

(2) During the epidemic period, national policies, scientific and technological investment and advancement, labor force, capital investment and other factors have been significantly adjusted, and the probability of various factors in the subsequent COVID-19 pandemic period will change. Is it reasonable to predict the economic growth in the later period based on the performance during the COVID-19 pandemic period? Please give the explanation in the paper.

Q2: Thanks for the reviewer's suggestion. The prediction of the MIDAS model is based on the number of prediction steps h. The number of prediction steps in this study is 12, which actually translates into quarterly data of 4 quarters, which is equal to one year. The accuracy of the MIDAS model is better for short-term forecasting, which also remains in the category of short-term forecasting. It is consistent with the time series prediction pattern and no extra-long prediction is made. In addition, as of now, the epidemic is still going on, so the model prediction is accurate. By comparing with the VAR model, we found that the MIDAS model has better accuracy in short-term forecasting, so we changed the number of forecasting steps to 3, which is equivalent to one quarter forward, in order to obtain more accurate forecasting results. The answer to this question is marked in line 328.

(3) Why didn't the author adopt one or more mainstream approach to predict economic growth? It is suggested that the main economic growth forecasting methods and empirical studies at home and abroad be summarized.

Q3: Thanks for the reviewer's suggestion. We reconstructed the VAR model as the base model for forecast comparison. The main domestic and foreign economic growth forecasting methods and empirical studies are summarized in the introduction. The forecast comparisons are labeled in line 312, and the summary forecast methods and empirical studies are labeled in line 60.

(4) Please explain the advantages of MIDAS and the forecasting idea adopted in this paper compared with other economic growth forecasting methods!

Q4: Thanks for the reviewer's suggestion. The MIDAS model balances the relationship between retaining valid information of high-frequency data and reducing the number of parameters to be estimated by con-structing distribution lag polynomials, thus effectively using high-frequency data to improve the prediction accuracy of low-frequency variables and avoiding a large loss of sample information. We added the advantages of MIDAS to the conclusion and labeled the predictive ideas of this paper in the article. The strengths are labeled in line 395 and the predictive ideas are labeled in line 190.

(5) There is a close relationship between carbon dioxide emissions and economic growth, which is generally a conclusion drawn in a long-term study. Some studies also point out that there is a nonlinear relationship between the two aspects. What is the basis for the author to believe that there must be a close relationship between the carbon dioxide emissions and economic growth? In addition to the research of some domestic and foreign authors, has the author of this manuscript conducted a similar pre-study? Please provide additional information on this point.

Q5: Thanks for the reviewer's suggestion. To test whether there is a correlation between carbon dioxide emissions and economic growth, we add a Granger causality test in the empirical part, and the results show that the two affect and constrain each other. Thus, there is a strong correlation between carbon emissions and economic growth. The answer to this question is marked in line 265.

(6) Could the author explain that there is an objective correlation between CO2 emissions and quarterly GDP based on the data simulation of 36 months in a three-year period in China? And now the epidemic situation is still continuing, it is suggested that the author update the data.

Q6: Thanks for the reviewer's suggestion. We added a Granger causality test in the empirical section, which con-firmed an objective correlation between the two. We updated the data to June 2022 and re-performed the empirical analysis. The answer to this question is marked in line 275.

(7) Why does the author not consider including the data before the epidemic? Why doesn't the author consider the monthly economic growth data? Please provide additional information on these two points.

Q7: Thanks for the reviewer's suggestion. GDP is the most authoritative indicator of economic development, and in China, the National Bureau of Statistics releases GDP data on a quarterly basis, and monthly data are not available. As for the carbon emission data before the epidemic, unfortunately, the data recorded in the data source of carbon emission starts from 2019. Therefore, we can only focus on studying the predicted impact of carbon emissions on economic growth during the epidemic, and cannot make a pre- and post-epidemic comparison. If extended data are subsequently obtained, pre- and post-epidemic comparisons can be further implemented in future studies. The answer to this question is marked in line 149.

(8) In the article, mainly discusses how to use monthly carbon dioxide emissions to forecast quarterly GDP. However, why does the author not consider that GDP fluctuations may cause changes in CO2 emissions? It is suggested to consider the two-way influence between the two.

Q8: Thanks for the reviewer's suggestion. We add Granger causality tests to the empirical evidence and find that they are mutually causal, and construct VAR models as the underlying models for comparison of prediction accuracy. The answer to this question is marked in line 270.

(9) It is suggested to include Chinese in the title.

Q9:Thanks for the reviewer's suggestion, Chinese has been added to the title.

(10) In addition, the change of CO2 emissions may also cause many environmental problems. To what extent will this affect economic growth? This is also a question to be discussed. It is suggested to consider it.

Q10: Thanks for the reviewer's suggestion. The previous development model achieved rapid economic development with high energy consumption and high pollution, which would not only increase the pressure of carbon emissions and aggravate the degree of environmental pollution, but also go against the sustainable development of economy. In addition, the environmental problems associated with carbon emissions, such as land drought, increased desertification and ocean acidification, can limit sustainable economic development in various ways. Therefore, a win-win situation of reducing carbon emissions and maintaining economic development can only be achieved by formulating appropriate carbon re-duction policies, such as implementing carbon neutral concepts in urban planning, and building environmentally friendly cities, livable urban communities and zero carbon buildings. We have added this to line 437.

Once again, we thank the reviewers for their efforts in this study.

Reviewer 3 Report

It is a very interesting research, the research problem is well explained and the authors make an interesting presentation of the controversial results found in the literature. The materials and methods are clearly  presented and leave no doubts fo how the objectives are achieved, the originality of the MIDAS model is well statated. Results are well presented and the discussion well founded. The theoretical contributiond are pointly summarized, as well as the policy implications. References are adequate and recent.

Author Response

Response to Reviewer 3 Comments

We thank the reviewers for their suggestions on this study. We have made the following main changes after receiving the review comments:

  1. Updated the latest data and redid the empirical analysis.
  2. Added Granger causality tests to demonstrate the correlation between carbon emissions and economic growth.
  3. Constructed a VAR model and used it as the base model for comparative forecasting.
  4. The number of forecasting periods is changed to 3 periods to obtain more accurate forecasting data.

Once again, we thank the reviewers for their efforts in this study.
